# Lactobacilli-Derived Postmetabolites Are Broad-Spectrum Inhibitors of Herpes Viruses In Vitro

**DOI:** 10.3390/ijms26010074

**Published:** 2024-12-25

**Authors:** Svetla Danova, Lili Dobreva, Kapka Mancheva, Georgi Atanasov, Lora Simeonova, Neli Vilhelmova-Ilieva

**Affiliations:** 1Department of General Microbiology, Stephan Angeloff Institute of Microbiology, Bulgarian Academy of Sciences, 26, Georgi Bonchev Str., 1113 Sofia, Bulgaria; stdanova@yahoo.com (S.D.); lili.ivailova1@gmail.com (L.D.); 2Institute of Biophysics and Biomedical Engineering, Bulgarian Academy of Sciences, 23, Georgi Bonchev Str., 1113 Sofia, Bulgaria; kapka_mancheva@abv.bg; 3Institute of Biodiversity and Ecosystem Research, Bulgarian Academy of Sciences, 25, Georgi Bonchev Str., 1113 Sofia, Bulgaria; gatanassov@gmail.com; 4Department of Virology, Stephan Angeloff Institute of Microbiology, Bulgarian Academy of Sciences, 26, Georgi Bonchev Str., 1113 Sofia, Bulgaria

**Keywords:** antiviral activity, herpes simplex virus type 1, Koi herpes virus, viral replication, virucidal effect, viral adsorption, *Lactobacillus probiotics*, postbiotics

## Abstract

Herpes viruses are highly contagious agents affecting all classes of vertebrates, thus causing serious health, social, and economic losses. Within the One Health concept, novel therapeutics are extensively studied for both veterinary and human control and management of the infection, but the optimal strategy has not been invented yet. Lactic acid bacteria are key components of the microbiome that are known to play a protective role against pathogens as one of the proposed mechanisms involves compounds released from their metabolic activity. Previously, we reported the anti-herpes effect of postmetabolites isolated from Lactobacilli, and here, we confirm the inhibitory properties of another nine products against the phylogenetically distant human Herpes simplex virus-1 (HSV-1) and fish Koi Herpes virus (KHV) in cell cultures. Cytotoxicity, cytopathic effect inhibition, virucidal effect, the influence on the adsorption stage of the virus to the cells, as well as the protective effect of the postmetabolites on healthy cells were evaluated. The inhibitory effect was more pronounced against HSV-1 than against KHV at all studied viral cycle stages. Regarding the intracellular replicative steps, samples S7, S8, and S9 (Mix group) isolated from *Ligilactobacillus salivarius* (vaginal strain) demonstrated the most distinct effect with calculated selective indices (SIs) in the range between 69.4 and 77.8 against HSV-1, and from 62.2 to 68.4 against KHV. Bioactive metabolites from various LAB species significantly inhibit extracellular HSV-1 and, to a lesser extent, KHV virions. The blockage of viral adsorption to the host cells was remarkable, as recorded by a decrease in the viral titer with Δlg ≥ 5 in the Mix group for both herpes viruses. The remaining postmetabolites also significantly inhibited viral adsorption to varying degrees with Δlg ≥ 3. Most metabolites also exerted a protective effect on healthy MDBK and CCB cells to subsequent experimental viral infection. Our results reveal new horizons for the application of LAB and their postbiotic products in the prevention and treatment of herpes diseases.

## 1. Introduction

Herpetic infections occur in all vertebrates with the severity and clinical manifestations of the disease depending both on viral and host factors [1,2]. According to the classification of the International Committee on Taxonomy of Viruses (ICTV) [3], the order *Herpesvirales* currently comprises three families: *Alloherpesviridae*, *Orthoherpesviridae*, and *Malacoherpesviridae*. *Alloherpesviridae* includes 12 species pathogenic to fish and amphibians, and one of them, which emerged in the late 1990s, is cyprinid herpesvirus 3, also called Koi herpes virus (KHV, CyHV-3) [4,5]. Since that time, KHV has rapidly spread over large areas causing significant economic losses in the carp aquaculture industry [6,7]. *Orthoherpesviridae* representatives infect reptiles, birds, and mammals [4]. There are eight human pathogens, of which human herpes virus type 1 (HSV-1) is the most common of all. According to the World Health Organization, as of 2024, approximately 3.8 billion people under the age of 50 years (64.2%) worldwide are infected with HSV-1 [8].

To date, various groups of nucleoside analogs that have been developed and approved for the treatment of HSV infections are used in practice [9,10,11,12,13,14,15]. They all work by interfering with viral DNA replication. The most widely administered is acyclovir—ACV {9-(2-hydroxyethoxymethyl) guanine}. In general, the major disadvantage of nucleoside analogue therapy is the emergence of viral drug resistance [16,17,18,19,20,21], which limits its clinical effectiveness, especially in immunocompromised patients [17,22,23]. This necessitates alternative, low-toxicity anti-herpes drugs with different targets of antiviral action to be developed and introduced into practice [22,23].

As is demonstrated by the literature, Lactobacilli are commensal residents of the gut with an essential role in human health. They produce substances such as lactic acid, H_2_O_2_, bacteriocins/BLIS (bacteriocin-like inhibitory compounds), exopolysaccharides, and others known as metabiotics, which can exert an antimicrobial effect against various pathogens, including viruses such as HSV [24,25]. The antiviral activity of lactic acid bacteria (LAB) might result from the production of metabolites, immune system modulation [24,26], and potential physical interference between bacterial cells and viral particles [24,27,28]. Dysbiosis and abnormal microbiota in Lactobacilli in the female vaginal tract have been associated with reduced protection against sexually transmitted diseases (STDs), including human immunodeficiency virus type 1 (HIV-1) and herpes simplex virus type 2 (HSV-2) [29,30].

Vaginal LAB are a significant factor in the prevention and treatment of HSV genital infection [31]. The exact mechanism of action toward genital herpes, however, has not been completely elucidated yet [31]. HSV-1 replication has been hindered in cell culture with LAB-derived cell-free supernatants (CSFs) added in the media [25]. Avitable et al., 2024, demonstrated the effectiveness of supernatants of two vaginal species (i.e., *Lactobaccillus crispatus* and *Lactobaccillus gasseri*) in neutralizing HSV-1 infection in two different cell lines of human and simian origin [25]. Fluid containing non-protein cell wall components from *Levilactobacillus brevis* cultivation has been shown to inhibit HSV-*2* multiplication [28].

Nine non-purified cell-free supernatants (CFSs) or whey fraction isolated from different strains of the genus *Lactobacillus* have been investigated for their antiviral effects at different stages of HSV-1 and KHV viral reproduction in the present study. We followed up the influence on extracellular virions, the interference with the adsorption to host cell receptors, and the intracellular replicative cycle of viruses, as well as the protective capacity of the pre-treatment of cells before the onset of viral infection.

## 2. Results

Prior to the antiviral experiments, the non-toxic range of each of the samples was first determined on Madin–Darby bovine kidney (MDBK) and common carp brain (CCB) cell lines permissive for viral replication. Cytotoxicity was recorded at the 48th hour on MDBK cells and at the 72th hour on CCB cultures, which is in accordance with the antiviral evaluation protocol (Figure 1).

The results in Table 1 reveal that all postbiotics are significantly more toxic than the reference chemotherapeutic acyclovir (ACV). Quite expectedly, the cytotoxicity was slightly higher in CCB cells, most likely because the exposure of the latter to the compounds lasted 24 h longer (72 h) than that of MDBK cells (48 h). In general, the Mix-sample group (S7, S8, and S9) had the greatest similar CC_50_ values corresponding to the lowest toxicity. S9 was minimally cytotoxic, followed by S6, S3, S1, and S4. The highest cytotoxicities (although in the order of the other samples) were demonstrated by S5 and S2 (Table 1).

The antiviral activity of Lactobacilli-derived metabolic products was determined at different stages of the viral cycle. The effect on the replication inside the cell was studied when the compounds were added after viral adsorption had expired (Figure 2).

Postmetabolites from the Mix group were the most potent against HSV-1 replication with SI = 77.8 calculated for S8, followed by S7 (SI = 73.7) and S9 (SI = 69.4). The strongest anti-KHV activity was shown by the same three products—S8 (SI = 68.4), S7 (SI = 65.7), and S9 (SI = 62.2). Among the others, the most distinct against HSV-1 replication was S5 (SI = 18.4), with similar activity, but S6 (17.8), S4 (SI = 17.3), and S3 (SI = 16.2) had slightly lower values of the selective indices. Similar trends were observed regarding the inhibition of KHV replication with the most active Mix group, followed by S5 (SI = 15.7), S6 (SI = 15.4), S4 (SI = 12.5), and S3 (SI = 12.0). By comparing postmetabolite capacities against the two viruses, we found that the inhibition of HSV-1 replication was slightly higher to that of KHV. S1 and S2 did not show any activity in regard to this stage of viral replication (Table 2).

The inhibitory effect of postmetabolites on extracellular herpes particles was tested in three different experimental setups. In the first panel, virions were placed in direct contact with each sample in a cell-free environment for various time intervals (5, 15, 30, 60, and 90 min) (Figure 3).

Initially, when HSV-1 was exposed to S3, S7, S8, and S9 for 5 min, the titers decreased only by Δlg = 1.25. The virucidal effect then increased proportionally to the incubation time. At 60 min of exposure, all samples except S1 and S2 significantly inhibited virions, as calculated for S3 and S7 with Δlg = 2.5; S4, S8, and S9 with Δlg = 2.0; and S5 and S6 with Δlg = 1.75 compared to viral control with no substance added. At the longest investigated time interval, some samples reached considerable virus-neutralizing action: S7 and S9 decreased titers by Δlg = 2.75, S8 by Δlg = 2.5, and S5 by Δlg = 2.0. If the virucidal effect of KHV virions is compared to that of HSV-1, the inhibition is significantly weaker. Only S7 showed an effect at 60 min of exposure with Δlg = 1.75, followed by S3, S4, S6, S8, and S9 with Δlg = 1.5 at the same time interval. Regarding KHV, only the Mix group showed a significant inhibitory effect: S7 (Δlg = 2.0), and S8 and S9 (Δlg = 1.75) at the longest time interval of 90 min. Samples S1 and S2 did not exert virucidal activity against both HSV-1 and KHV virions at any of the incubation times studied (Table 3).

In the second experimental design, the investigated postmetabolites were incubated together with the viral particles at the time of their adsorption to the sensitive cells. The inhibitory effect on virus adsorption was also monitored at different time intervals (15, 30, 45, and 60 min) (Figure 4).

Four of the postmetabolites showed remarkable inhibition of HSV-1 adsorption even at the shortest time interval of 15 min. The effect was most pronounced in S5 with a decrease in viral titer with Δlg = 2.5, followed by S3 (Δlg = 2.25), and S4 and S6 (Δlg = 2.0). The Mix group demonstrated weak effects initially (Δlg = 1.25), but as the exposure time was prolonged, their activity increased sharply at each subsequent time interval, and at the 60th min, they had greatest values: S7 (Δlg = 5.75), S8, and S9 (Δlg = 5.5). All other samples also significantly blocked HSV-1 adsorption at 60 min as follows: S5 (Δlg = 4.5), S4 and S6 (Δlg = 3.75), and S2 and S3 (Δlg = 3.5).

The adsorption of KHV also appeared to be dependent on the incubation time. At 15 min after exposure, the strongest effect was observed for S5 (Δlg = 2.0), followed by S3, S4, and S6 (Δlg = 2.0), then S2 (Δlg = 1.75). The samples from the Mix group showed negligible activity at Δlg ≤ 1.0. As the exposure time increased, their activity considerably improved, and at the 60th min, they lowered the viral titer by Δlg ≥ 5. The activity of the other postbiotics also increased with the exposure time and, at 60 min, reached Δlg = 4.0 (at S5), Δlg = 3.75 (at S2), Δlg = 3.5 (at S6), and Δlg = 3.0 (at S3 and S4). The inhibitory effect of S1 on the adsorption of both herpes viruses was weak, being slightly stronger for HSV-1 (Table 4).

By the third approach, we verified whether postmetabolites could protect healthy cells from subsequent herpes infection. The susceptible cells were placed in contact only with the tested compounds once again for various time intervals (5, 15, 30, 60, and 120 min) before the experimental viral challenge (Figure 5).

After 5 min of incubation of the MDBK cells, a protective effect was shown by the Mix group: S8 and S9 (Δlg = 2.0), and S7 (Δlg = 1.75), as well as S4 with Δlg = 2.0. The longer the postmetabolites were incubated with the intact cells, the stronger the protective effect they had on them. At 30 min of exposure, almost all postmetabolites had a significant influence on the severity of the subsequent infection. The most significant effect was seen for S8 (Δlg = 2.75), followed by the other representatives of the Mix group, S7 and S9 (Δlg = 2.5). Up to 120 min of exposure, all postmetabolites increased their activity. Again, the influence of the Mix group, with a decrease in the viral titer by Δlg ≥ 3 and the remaining samples with Δlg = 2.5 (S3 and S6) and Δlg = 2.25 (S4 and S5), was most pronounced.

The protective effect on CCB cells was slightly lower than that on MDBK cells. At 5 min of exposure, the Mix group and S4 reduced viral titers by Δlg = 1.75. For all postmetabolites, once again, the influence increased with the exposure time. At the 120th min, the most distinct effect was seen for S9 (Δlg = 3.0), followed by S8 (Δlg = 2.75) and S7 (Δlg = 2.5). Four other postmetabolites demonstrated a protective effect of Δlg = 2.0 (S3–S6). S1 and S2 did not have a significant protective effect on MDBK and CCB cells, and even at the longest time interval, viral titer reduction was only Δlg ≤ 1.25 (Table 5).

## 3. Discussion

A number of studies report the antiviral effects of probiotics and data support their role in the development of therapeutics [33,34]. Several possible mechanisms have been suggested to explain their beneficial effects in the infection state. The interaction between bacterial cells and host factors, direct interference of probiotic strains with viral particles, neutralization of viral particles by probiotic metabolites such as protein molecules or organic acids, and/or stimulation of the immune system could work individually or together to result in antiviral effects [26,35]. Particular inhibitory compounds have not been specified in the reports, but the effect appears to be dependent on both the type of probiotic and the virus. Viral suppression by vaginal Lactobacilli can be a result of aggregation of viral particles and/or the action of released antimicrobials [26]. Lactobacilli release various metabolic products such as lactic acid, hydrogen peroxide, bacteriocins, and exopolysaccharides, which exhibit antibacterial, antifungal, and antiviral activity [36,37,38,39,40,41,42]. H_2_O_2_ is one of the metabolites produced by *L. acidophilus* that inhibits many viruses including HIV [43]. Lactic acid has been shown to be a major antiviral factor produced by Lactobacilli in the vaginal mucosa [44,45]. Moreover, HSV-2 is irreversibly inactivated by lactic acid in the vagina of a healthy human [31,46]. Lactic acid was found to be essential for the denaturation of the viral capsid [47]. Although the antibacterial activity of bacteriocins has been partially described, their antiviral mechanism has not been fully understood yet. It is suggested that bacteriocins can lead to the aggregation of viral particles, block viral particles by binding to host cell receptors, or inhibit later key steps in the viral replication cycle [48]. A bacteriocin produced by *Enterococcus faecium* ST5Ha can block HSV-1 without exhibiting toxicity to cells [49]. An anti-HSV effect was also shown by trappin-2 and elafin, a serine protease inhibitor [50]. Relevant studies have confirmed that probiotics can secrete extracellular proteins, weaken pathogen adhesion, and protect intestinal cells [51]. Supernatants from various Lactobacilli can inhibit HSV replication [31,52].

LAB can also produce extracellular vesicles (EVs). EVs inhibit HIV-1 infection in vitro by reducing HIV-1 entry/attachment to target cells [30]. Besides producing metabolites, probiotics can inhibit virions directly by interacting with them or competing for cellular receptors, thus preventing virus entry into host cells [53].

Our results are consistent with other research data and correlate with our previous work, where we proved the antiherpetic activity of postmetabolites isolated from LAB strains [32,54]. Most of the postmetabolites investigated in the present study demonstrated an inhibitory effect on enveloped extracellular herpesvirus particles, the effect being notably stronger for HSV-1. Applied simultaneously with viral particles during the attachment, the studied postmetabolites distinctly affected the stage of viral adsorption to sensitive cells, and the action of some of the postmetabolites led to a decrease in viral titers by more than lg > 4–5.

During the preliminary treatment of the healthy MDBK and CCB cells with the studied bacterial products, we recorded protection from subsequent herpes infection, expressed as a decrease in viral titers with ∆lg ≥ 2–3. Similarly, when cells were pre-treated with metabolites isolated from *L. plantarum* strain N4, significant protection from subsequent infection with gastroenteritis coronavirus was observed [55]. A possible mechanism of action that can explain this effect is an influence of probiotic bacteriocins on cellular receptors responsible for recognizing and binding the virus [48].

Data suggest that probiotics may also affect the intracellular stages of the viral cycle [56], which is consistent with our results, showing the postmetabolites are selective inhibitors according to SI values obtained, both against HCV-1 and against KHV replication (Table 2). The anti-HSV activity of ASP2151 has been demonstrated by targeting the helicase–primase complex of HSV-1 and HSV-2 [57]. Antiviral activity against HSV-1 [54] is exhibited by enterosin ST5Ha (ST5Ha). In Vero and BHK-21 cells, the antiviral activity of enterosin ST4V and CRL35 against herpes simplex thymidine kinase type 1 and 2 positive and defective strains was found to affect intracellular viral proliferation and inhibit the late phases of the cycle [48,58,59].

The presented results constitute just the initial stage of our research, which needs to further establish the detailed chemical composition of the products containing postmetabolites, to verify the individual effect of each constituent on the viral replicative cycle, and to clarify the exact anti-herpes mechanism in the cell.

## 4. Materials and Methods

### 4.1. Lactobacilli and Postmetabolites Tested

Selected Lactobacilli from the Collection of Lactic Acid Bacteria and Probiotics Laboratory, Stephan Angeloff Institute of Microbiology, were used. They are representatives of different species and were isolated from various habitats (Table 6). LAB strains were stored at −20 °C in MRS broth supplemented with 20% *v*/*v* glycerol. For experimental purposes, each strain was pre-cultured twice in MRS broth (HiMedia, Mumbai, India), to obtain a high number of living cells synchronized in terms of their growth phase and metabolic activity. Thus, after storage at (−20 °C) in a medium supplemented with glycerol (20% *v*/*v*), two passages in fresh MRS broth were made. A 5% inoculum (*v*/*v*) from a *Lactobacillus* stock was used in the first culture for each of the selected strains. They were incubated at 37 °C and late exponential phase cultures were obtained, standardized by MacFarland 4.0 standards (~1.2 × 10^9^ CFU/mL) and used as a 5% (*v*/*v*) inoculum for the second cultures at 37 °C in incubator Nuve EN 400 (Nuve, Ankara, Turkey). The cell-free supernatants (CFSs) from the second cultures were collected by centrifugation (3000 rcf, 5 min, centrifuge Hermle Labortechnik GmbH, Wehingen, Germany) and were used for the in vitro test. Collected CFSs (named postmetabolites), from late-exponential-(24 h) or stationary-(48 h) phase LAB cultures in De Man–Rogosa–Sharpe broth (pH 6.5) were filtered with 0.22 µm MF-Millipore™ syringe filters (Merck KGaA, Darmstadt, Germany) and stored at −20 °C prior to assessment for anti-herpes virus activity. In order to simulate a multibacterial probiotic formula, 4 *Lactobacillus* strains were combined as 4 pre-cultures and were inoculated (1.25% *v*/*v* from each strain) in MRS broth (HiMedia, Mumbai, India). Obtained CFSs after 48 h of cultivation at 37 °C were assessed.

Harvested cells from the second exponential *Lactobacillus* cultures, washed with sterile PBS (pH 6.0), were used as an inoculum (10% *v*/*v*) in sterile reconstituted (10% *w*/*v*) skimmed milk. Milk fermented overnight at 37 °C with mixed four *Lactobacillus* strains were centrifuged at 1500 rcf 5 min, (Hermle, Wehingen, Germany).

### 4.2. Cells

Madin–Darby bovine kidney (MDBK) cells were obtained from the National Bank for Industrial Microorganisms and Cell Cultures, Sofia. The cell lines were grown in Dulbecco’s modified Eagles’s medium (DMEM) containing 10% fetal bovine serum (GibcoTM, Waltham, MA, USA), supplemented with 10 mM HEPES buffer (Merck, Darmstadt, Germany) and antibiotics (penicillin 100 IU/mL, streptomycin 100 µg/mL) in a CO_2_ incubator (HERA cell 150, Heraeus, Hanau Germany) at 37 °C/5% CO_2_.

The European Union Reference Laboratory for Fish and Crustacean Diseases, National Institute of Aquatic Resources, Technical University of Denmark, kindly provided the common carp brain (CCB) cell line. The cells were grown in EMEM medium + 2 mM glutamine + 1% non-essential amino acids (NEAAs) + 10% fetal bovine serum (Gibco BRL, Paisley, UK) with 10 mM HEPES buffer (Merck, Germany) and antibiotics (penicillin 100 IU/mL, streptomycin 100 μg/mL) in a CO_2_ incubator (HERA cell 150, Heraeus, Germany) at 22 °C/5% CO_2_.

### 4.3. Viruses

Herpes simplex virus type 1 (HSV-1), Victoria strain, was received from Prof. S. Dundarov, the National Center of Infectious and Parasitic Diseases, Sofia. The virus was replicated in a confluent monolayer of MDBK cells in a maintenance solution of DMEM (Gibco BRL, Paisley, UK), plus 0.5% fetal bovine serum (Gibco BRL, UK). The infectious titer of the stock virus was 10^9.0^ CCID_50_/mL.

Koi herpes virus (KHV), F-347 strain, was purchased from the American Type Culture Collection (ATCC, Manassas, VA, USA). The virus was replicated in monolayer CCB cells in a maintenance solution of EMEM (Gibco BRL, Paisley, UK), plus 0.5% fetal bovine serum (Gibco BRL, Paisley, UK), in a CO_2_ incubator at 22 °C/5% CO_2_. The infectious titer of the stock virus was 10^6.25^ CCID_50_/_mL_.

### 4.4. Cytotoxicity Assay

A confluent monolayer cell culture in 96-well plates (Costar^®^, Corning Inc., Kennebunk, ME, USA) was treated with a 0.1 mL/well supported medium containing decreasing concentrations of test extracts. The cells were incubated at 37 °C and 5% CO_2_ for 48 (for MDBK cells) or 72 h (for CCB cells). After microscopic evaluation, the medium containing the test sample was removed and the cells were washed with PBS and incubated with neutral red at 37 °C for 3 h. The neutral uptake assay used has been described in detail in our previous studies [32,54]. The 50% cytotoxic concentration of material that reduced cell viability by 50% compared to untreated controls (CC_50_) was determined. The maximum tolerated concentration (MTC) of the extracts at which they did not affect the cell monolayer was also determined. Each sample was tested in triplicate, with four cell culture wells per test sample.

### 4.5. Determination of Infectious Viral Titers

Cells were cultured in 96-well plates. After monolayer formation, cells were infected with 0.1 mL of virus suspension in tenfold falling dilutions by the terminal dilution method [63]. After viral adsorption, the unabsorbed virus was removed and 0.1 mL/well of support medium was added to the cells. The plates were incubated at the appropriate temperature for the given strain (37 °C for HSV-1 and 22 °C for KHV) in a CO_2_ incubator HERA cell 150 (Radobio Scientific Co., Ltd., Shanghai, China) for 2 (for HSV-1) or 3 (for KHV) days. Cells infected with a maximal concentration of virus and demonstrating a maximal cytopathic effect were used as a control. Infectious viral titer was determined by microscopic observation of the cell monolayer and cytopathic effect (CPE) determination. Visually determined CPE was confirmed by staining with the neutral red (NR) dye (neutral red uptake test) as described in Sect. Viral titers are expressed as lg IU (infectious units) cell culture infectious dose of 50% (CCID_50_)/1 mL.

### 4.6. Effect on Viral Replication

The antiviral activity of the tested postmetabolites was determined by the cytopathic effect (CPE) inhibition test. A confluent cell monolayer in 96-well plates was infected with an infectious dose of 100 CCID_50_/1 mL. After 60 min of viral adsorption, the unabsorbed virus was discarded and the test compound was added at various concentrations; cells were incubated for 48 h at 37 °C and 5% CO_2_ for HSV-1 or 72 h at 22 °C and 5% CO_2_ for KHV. The cytopathic effect was determined using a neutral red uptake assay, and the percentage of CPE inhibition for each concentration of the tested sample was calculated using the following formula:% CPE = [OD_test sample_ − OD_virus control_]/[OD_toxicity control_ − OD_virus control_] × 100
where OD_test sample_ is the mean value of the ODs of the wells inoculated with the virus and treated with the test sample in the respective concentrations; OD_virus control_ is the mean value of the ODs of the virus control wells (with no compound in the medium); and OD_toxicity control_ is the mean value of the ODs of the wells not inoculated with the virus, but treated with the corresponding concentration of the test compound. The 50% inhibitory concentration (IC_50_) was defined as the concentration of the test substance that inhibited 50% of viral replication when compared to the virus control. The selectivity index (SI) was calculated from the ratio CC_50_/IC_50._

### 4.7. Virucidal Assay

Samples containing the virus from the starting stock (described in Section 4.3) and a tested postmetabolite at its maximum tolerated concentration (MTC) were combined in a 1:1 ratio and subsequently stored at room temperature for various time intervals (5, 15, 30, 60, and 120 min). Ethanol 70% was used as a control. Residual infectious viral content in each sample was then determined by the end-point dilution method, and ∆lgs compared to untreated controls were estimated.

### 4.8. Effect on Viral Adsorption

A monolayer of the respective cell cultures seeded in 24-well plates were pre-cooled to 4 °C and inoculated with 10^6^ CCID_50_ of the respective viral strain. In parallel, they were treated with tested extracts at their MTC and incubated at 4 °C for the time of viral adsorption. At various time intervals (15, 30, 45, and 60 min), cells were washed with PBS to remove both the substance and unattached virus; cells were then covered with maintenance medium and incubated at 37 °C and 5% CO_2_ for HSV-1 or at 22 °C and 5% CO_2_ for KHV for 24 h. After freezing and thawing three times, the infectious virus titer of each sample was determined by the end-point dilution method. ∆lgs was determined relative to the viral control (no postmetabolite added). Each sample was prepared in quadruplicate.

### 4.9. Pre-Treatment of Healthy Cells

MDBK and CCB cell monolayers grown in 24-well cell culture plates (CELLSTAR, Greiner BioOne, Kremsmünster, Austria) were treated for various time intervals (5, 15, 30, 60, and 120 min) with the MTC of the tested postmetabolite in the maintenance medium (1 mL/pit). After the above time intervals, postmetabolites were removed and cells were washed with phosphate buffered saline (PBS) and inoculated with the respective viral strain (1000 CCID_50_ in 1 mL/well). After 60 min of adsorption, the unattached virus was discarded and the cells were covered with the maintenance medium. Samples were incubated at 37 °C and 5% CO_2_ for HSV-1 or at 22 °C and 5% CO_2_ for KHV for 24 h, and after freezing and thawing three times, infectious viral titers were determined by the terminal dilution method. ∆lg was calculated based on the viral control (no compound) and compound-treated group value subtraction. Each sample was prepared in quadruplicate.

### 4.10. Determination of Protein Content (mg/mL) in the Tested Samples

The protein content of the samples was determined with a commercial HiMedia Bradford kit (HiGenoMB, HiMedia, India), according to the manufacturer’s instructions. A standard curve with BSA (bovine serum albumin; mg/mL) was built, and the concentration was expressed in mg/mL.

### 4.11. Statistical Analysis

Cytotoxicity (CC_50_) and antiviral effects (IC_50_) were presented as means ± SD. The differences between the cytotoxicity values of postmetabolites and ACV, as well as between the effects of the test products on viral replication were assessed by Student’s *t*-test, and *p*-values of <0.05 were considered significant. The final data sets were analyzed statistically using Graph Pad Prism 4 program.

## 5. Conclusions

We evaluated the antiviral properties of metabolic products isolated from different strains of Lactobacilli against HSV-1 and KHV on intracellular and extracellular stages of viral replication in vitro. The inhibition of virions outside of the cell, blockage of the adsorption to host cells, as well as the protective effect on healthy cells suggests that LABs and their derivatives could be considered as a supplement or/alternative treatment approach for optimized control of herpes infections and their recurrences. Despite the incontrovertible inhibitory effect reported, the exact mechanism underlying the activity of probiotics and their postmetabolites against herpes viruses remains to be clarified. This opens up new horizons in the research of probiotics and their postmetabolites for the development of novel therapeutics of viral infections.

## Figures and Tables

**Figure 1 ijms-26-00074-f001:**
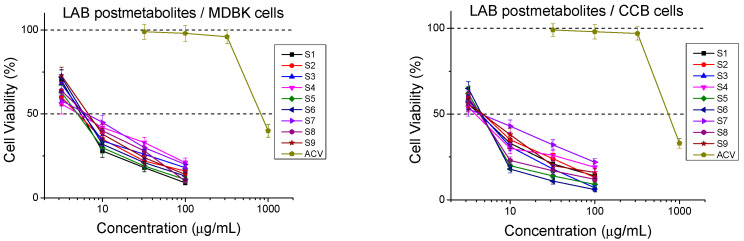
Cytotoxic effect of postmetabolites.

**Figure 2 ijms-26-00074-f002:**
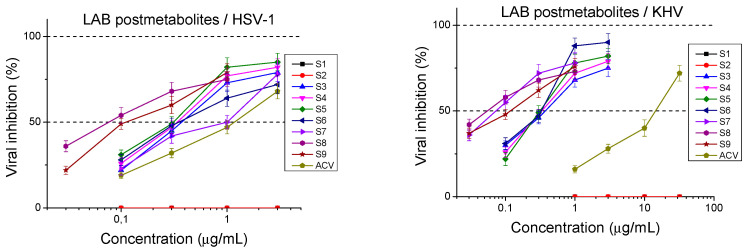
Inhibitory effect of postmetabolites on the viral replicative cycle.

**Figure 3 ijms-26-00074-f003:**
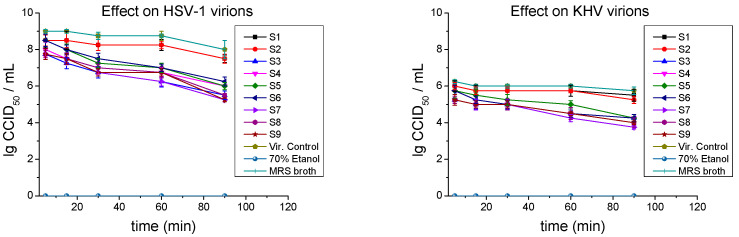
Inhibitory effect of LAB postmetabolites on extracellular virions.

**Figure 4 ijms-26-00074-f004:**
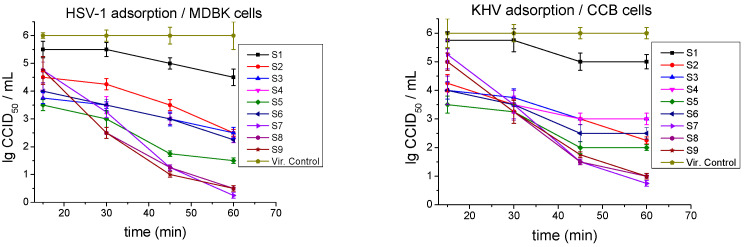
Influence of postmetabolites on the viral adsorption stage.

**Figure 5 ijms-26-00074-f005:**
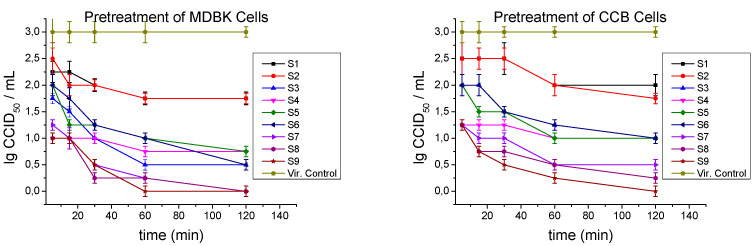
Protective effect of postmetabolites on healthy cells before the onset of viral infection.

**Table 1 ijms-26-00074-t001:** Cytotoxicity of LAB postmetabolites in vitro against MDBK and CCB cell lines.

Tested Samples	MDBK Cell Line		CCB Cell Line	
	CC_50_ (µg/mL)	MTC (µg/mL)	CC_50_ (µg/mL)	MTC (µg/mL)
S1	6.3 ± 0.6 ***	1.0	4.8 ± 0.4 ***	1.0
S2	5.8 ± 0.6 ***	1.0	5.5 ± 0.6 ***	1.0
S3	6.7 ± 0.6 ***	1.0	5.2 ± 0.6 ***	1.0
S4	6.2 ± 0.7 ***	1.0	4.5 ± 0.7 ***	1.0
S5	5.9 ± 0.4 ***	1.0	5.2 ± 0.4 ***	1.0
S6	6.9 ± 0.6 ***	1.0	5.3 ± 0.6 ***	1.0
S7	7.4 ± 0.8 ***	1.0	5.2 ± 0.8 ***	1.0
S8	7.1 ± 0.5 ***	1.0	5.0 ± 0.5 ***	1.0
S9	7.6 ± 0.4 ***	1.0	5.3 ± 0.4 ***	1.0
MRS broth	-	-	-	-
ACV	873.4 ± 6.8	320.0	820.0 ^	320.0 ^

*** *p* < 0.0001, comparing each postmetabolite with ACV; Student’s *t*-test. ^ the result was presented in a previous study [32].

**Table 2 ijms-26-00074-t002:** Antiviral activity of LAB postmetabolites against the viral replication cycle of herpes viruses in vitro.

Tested Samples	HSV-1		KHV	
	IC_50_ (µg/mL)	SI	IC_50_ (µg/mL)	SI
S1	-	-	-	-
S2	-	-	-	-
S3	0.42 ± 0.08 ***	16.2	0.43 ± 0.06 ***	12.0
S4	0.36 ± 0.06 ***	17.3	0.36 ± 0.02 ***	12.5
S5	0.32 ± 0.04 ***	18.4	0.33 ± 0.04 ***	15.7
S6	0.39 ± 0.05 ***	17.8	0.34 ± 0.05 ***	15.4
S7	0.1 ± 0.01 ***	73.7	0.08 ± 0.004 ***	65.7
S8	0.09 ± 0.006 ***	77.8	0.07 ± 0.002 ***	68.4
S9	0.11 ± 0.02 ***	69.4	0.12 ± 0.009 ***	62.2
MRS broth	-	-	-	-
ACV	1.2 ± 0.8	727.8	16.2 ± 2.3 ^	50.6 ^

*** *p* < 0.0001, comparing each postmetabolite with ACV; Student’s *t*-test. ^ the result was presented in a previous study [32].

**Table 3 ijms-26-00074-t003:** Virucidal activity of LAB postmetabolites against HSV-1 and KHV virions.

Tested Samples	Δlg									
			HSV-1					KHV		
	5 min	15 min	30 min	60 min	90 min	5 min	15 min	30 min	60 min	90 min
S1	0.5	0.5	0.5	0.5	0.5	0.25	0.25	0.25	0.25	0.25
S2	0.5	0.5	0.5	0.5	0.5	0.25	0.25	0.25	0.25	0.5
S3	1.25	1.75	2.0	2.5	2.5	1.0	1.0	1.0	1.5	1.5
S4	1.0	1.5	1.75	2.0	2.0	1.0	1.0	1.0	1.5	1.5
S5	0.5	1.0	1.5	1.75	2.0	0.5	0.5	0.75	1.0	1.5
S6	0.5	1.0	1.25	1.75	1.75	0.5	0.75	1.0	1.5	1.5
S7	1.25	1.5	2.0	2.5	2.75	1.0	1.0	1.0	1.75	2.0
S8	1.25	1.5	1.75	2.0	2.5	1.0	1.0	1.0	1.5	1.75
S9	1.25	1.5	2.0	2.0	2.75	1.0	1.0	1.0	1.5	1.75
MRS broth	0.0	0.0	0.0	0.0	0.0	0.0	0.0	0.0	0.0	0.0
70% ethanol	9.0	9.0	8.75	8.75	8.0	6.25	6.0	6.0	6.0	5.75

**Table 4 ijms-26-00074-t004:** Effect of LAB postmetabolites on the step of viral adsorption to susceptible cells.

Tested Samples	Δlg							
	HSV-1				KHV			
	15 min	30 min	45 min	60 min	15 min	30 min	45 min	60 min
S1	0.5	0.5	1.0	1.5	0.25	0.25	1.0	1.0
S2	1.5	1.75	2.5	3.5	1.75	2.5	3.0	3.75
S3	2.25	2.5	3.0	3.5	2.0	2.25	3.0	3.0
S4	2.0	2.5	3.0	3.75	2.0	2.5	3.0	3.0
S5	2.5	3.0	4.25	4.5	2.5	2.75	4.0	4.0
S6	2.0	2.5	3.0	3.75	2.0	2.5	3.5	3.5
S7	1.25	2.75	4.75	5.75	0.75	2.5	4.5	5.25
S8	1.25	3.5	4.75	5.5	1.0	2.75	4.5	5.0
S9	1.25	3.5	5.0	5.5	1.0	2.75	4.25	5.0

**Table 5 ijms-26-00074-t005:** Pre-treatment of MDBK and CCB cells with LAB-derived fragments or postmetabolites before viral infection.

Tested Samples	Δlg									
	HSV-1							KHV		
	5 min	15 min	30 min	60 min	120 min	5 min	15 min	30 min	60 min	120 min
S1	0.75	0.75	1.0	1.25	1.25	0.5	0.5	0.5	1.0	1.0
S2	0.5	1.0	1.0	1.25	1.25	0.5	0.5	0.5	1.0	1.25
S3	1.25	1.5	2.0	2.5	2.5	1.0	1.0	1.5	1.75	2.0
S4	2.0	2.0	2.0	2.25	2.25	1.75	1.75	1.75	2.0	2.0
S5	1.0	1.75	1.75	2.0	2.25	1.0	1.5	1.5	2.0	2.0
S6	1.0	1.25	1.75	2.0	2.5	1.0	1.0	1.5	1.75	2.0
S7	1.75	2.0	2.5	2.75	3.0	1.75	2.0	2.0	2.5	2.5
S8	2.0	2.0	2.75	2.75	3.0	1.75	2.25	2.25	2.5	2.75
S9	2.0	2.0	2.5	3.0	3.0	1.75	2.25	2.5	2.75	3.0

**Table 6 ijms-26-00074-t006:** LAB samples tested in the present work.

Tested Samples	Species and Strain	Origin
S1	*Lactiplantibacillus plantarum L5*	Homemade sample of katak [60]
S2	*Lactiplantibacillus plantarum L6*	[61]
S3	*Lactiplantibacillus plantarum KCC1*	Artisanal white brined cheese
S4	*Lactiplantibacillus plantarum RL34*	[62]
S5	*Ligilactobacillus salivarius* S1	Vaginal strain [61]
S6	*Lactiplantibacillus paraplantarum 7S*	Breast milk
S7	*Mix LS-48 h MRS*	
S8	*Mix LS-WF*	
S9	*Mix 8*	

## Data Availability

The raw data supporting the conclusions of this article will be made available by the authors on request.

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
