# Peer review of "Lactobacilli-Derived Postmetabolites Are Broad-Spectrum Inhibitors of Herpes Viruses In Vitro"

_ijms, 2024, doi:10.3390/ijms26010074_

Round 1

Reviewer 1 Report

Comments and Suggestions for Authors The description of the materials and methods, and presentation and description of results must be improved. The English of the publication must be improved by a native English speaker.   For Latin names use Italic. Lines 44-47 – quote, what about the year of reference? Check and correct the taxonomy of Herpesvirales. Lines 96-97 “The results in Table 4 reveal that all postbiotics are significantly less toxic than the reference chemotherapeutic acyclovir (ACV).” Do you mean Table 1? The data of Table 1 show ACV was more than 100 times less cytotoxic than postmetabolites, and whereas you claim the opposite. For example 873.4/6.3>100.   Reviseand correct the titles of all tables Tables 1 and 2 - mL but not ml You have the liquid - a mixture of postmetabolites. How do you convert it to mg? Table 2 – improve the title of Table.  

Table 3 Does the data presented in Table mean that 70% ethanol could inactivate all (100%) or zero (0%) HSV-1 in 5 minutes, while in 30 min 43.7% or 56.3%?

How do you explain that the titre of virus control and ethanol 70%-virus mixture showed identical titre? For example, the titre 109 ID50/mL means 1000000000 ID50/mL. When you prepare 1mL ethanol 70%-HSV-1 mixture you have to take about 0.77 mL of 95% ethanol and 0.25 mL of HSV-1 109 ID50/mL. Subsequently, 1 mL of 70%-HSV-1 mixture contains 108.4 ID50/mL.

When you prepare 1mL ethanol 70%-KHV mixture you have to take about 0.77 mL of 95% ethanol and 0.25 mL of KHV106.25 ID50/mL. Subsequently, 1 mL of 70%-KHV mixture contains 105.65 ID50/mL.

Explain how you prepared ethanol 70%-virus mixtures. Explain why dilution of viruses with ethanol did not result in the expected log reduction of virus titre.

Table 4

You used samples containing virus (105 CCID50) (lines 342), and found virus log reduction when treating with for S7, S8, and S9, respectively - 5.75 log, 5.5 log, and 5.5 log for HSV1, and 5.25 log , 5.0 and 5.0 and KHV respectively. Explain what were the titres of virus controls after incubation.

Table 5. Pretreatment of MDBK cells with LAB-derived fragments or postmetabolites before viral infection.

The title? Did not you pretreat CCB for KHV? What were the titres of virus controls ?

Materials and Methods (4.1. Lactobacilli and post-metabolites tested:)

Explain how each strain was pre-cultured twice. How many mL of MRS broth was used, what was the temperature for the cultivation, and what was the bacteria dose for the first and the second pre-culturing? Was the dose the same for every strain? Was the number of bacteria determined after the first and the second preculturing?

Lines 262-263 and 28-270 Compare quotes “In order to simulate a multibacterial probiotic formula 4 Lactobacillus strains were combined and mixed CFS were assessed.“ and “Fermented over night milk with a mix of four strains were centrifuged at 1500 rcf 5 min, (Hermle- Germany centrifuge) and the whey fraction (Mix LS-WF) was collected for the assay.” At first, I understood that Lactobacillus strains were not mixed and were not grown together,- and that only CFS of separate strains were mixed, and later it became unclear if milk with a mix of four strains was fermented or not.

4.4. Cytotoxicity Assay

Line 302 check the incubation time 96 or 120 hours?

Table Lactiplantibacillus but not Lactiplantibaillus

Choose postmetabolites or post-metabolites

4.5. Determination of Infectious Viral Titers

Lines 320-321 Quote“Viral titers are expressed as lg IU (infectious units) cell culture infectious dose of 50% (CCID50) / 0.1 ml.” 0.1 mL or 1 ml?

4.6. Antiviral Activity Assay

Change the title of the section since 4.7, 4.8, and 4.8 mean testing of antiviral activity.

4.7. Virucidal Assay

Quote “Samples containing virus (105 CCID50) and tested postmetabolite at its maximum tolerated concentration (MTC) were combined in a 1:1 ratio”. What was the volume? Explain more in detail. What did the volume of the sample contained 105 CCID50?

70% ethanol control is not described.

4.9. Pre-Treatment of Healthy Cells

What about pretreatment of CCB for KHV testing?

Author Response

Comments 1: The description of the materials and methods, and presentation and description of results must be improved.

Response 1: Additional information is included in both the materials and methods and the results.

Comments 2: The English of the publication must be improved by a native English speaker.  

Response 2: The English in the manuscript was improved by an English-speaking colleague.

Comments 3: For Latin names use Italic.

Response 3: All Latin names are in Italics. The exception is the names of viruses, because they are not italicized at all - only the systematic groups to which they belong.

Comments 4: Lines 44-47 – quote, what about the year of reference? Check and correct the taxonomy of Herpesvirales

Response 4: Reference number 4 is from 2029. But we do not cite the classification of herpes viruses with it. From reference number 3 we have taken the latest classification, which is cited on line 44.

Comments 5: Lines 96-97 “The results in Table 4 reveal that all postbiotics are significantly less toxic than the reference chemotherapeutic acyclovir (ACV).” Do you mean Table 1? The data of Table 1 show ACV was more than 100 times less cytotoxic than postmetabolites, and whereas you claim the opposite. For example 873.4/6.3>100. 

Response 5: We thank the esteemed reviewer for the comments made on Table 1. We have corrected the article number. We have also corrected the statement about the cytotoxicity of postmetabolites. The results presented in Table 2 show that it is precisely from the higher cytotoxicity and lower IC50 values ​​that significantly lower selectivity indices of the test products are obtained.

Comments 6: Revise and correct the titles of all tables Tables 1 and 2 - mL but not ml 

Response 6: The appropriate corrections have been made.

Comments 7: You have the liquid - a mixture of postmetabolites. How do you convert it to mg?

Response 7: Тhe postmetabolites produced during the fermentation contains different type of molecules, such as peptides/proteins, low molecular mass and bacteriocin-like substances, SCFA (lactat, butirat, etc.). We determined the protein/peptide contain by classical Bradford method. The results from Bradford assay (using a Bradford assay kit, HiMedia) are presented in mg/mL for each of tested variants. The details for protein content measurement was added in edited variant оf the manuscript. A new paragraph for protein estimation was added as 4.10 as follow:

            New section 4.10: The protein content of the samples was determined with commercial HiMedia Bradford kit (HiGenoMB, HiMedia, Maharashtra, India), according to the manufacturer’s instructions. A standard curve with BSA (Bovine serum albumin; mg/mL) was built, and the concentration was expressed in mg/mL.

 Comments 8: Table 2 – improve the title of Table.  

Response 8: The title of Table 2 has been changed.

Comments 9: Table 3 Does the data presented in Table mean that 70% ethanol could inactivate all (100%) or zero (0%) HSV-1 in 5 minutes, while in 30 min 43.7% or 56.3%? How do you explain that the titre of virus control and ethanol 70%-virus mixture showed identical titre? For example, the titre 109 ID50/mL means 1000000000 ID50/mL. When you prepare 1mL ethanol 70%-HSV-1 mixture you have to take about 0.77 mL of 95% ethanol and 0.25 mL of HSV-1 109 ID50/mL. Subsequently, 1 mL of 70%-HSV-1 mixture contains 108.4 ID50/mL. When you prepare 1mL ethanol 70%-KHV mixture you have to take about 0.77 mL of 95% ethanol and 0.25 mL of KHV106.25 ID50/mL. Subsequently, 1 mL of 70%-KHV mixture contains 105.65 ID50/mL. Explain how you prepared ethanol 70%-virus mixtures. Explain why dilution of viruses with ethanol did not result in the expected log reduction of virus titre.

Response 9: Ethanol 70% inactivated 100% of all HSV-1 and KHV virions. This was the result in all tested intervals of time. The slight decrease in Δlg with increasing exposure time is due to the fact that the titers of the viral controls themselves decreased over time, but the activity of ethanol remained 100%. The mixtures themselves were prepared in 1 ml. 760 µL of ethanol (96%) and 240 µL of the tested sample. Indeed, according to the calculations, the HSV-1 control should contain 108.4 ID50/mL, and the KHV control should contain 105.65 ID50/mL, but when analyzing the results, the viral controls demonstrate viral titer as high as the titer of the starting stock of the virus.

Comments 10: Table 4

You used samples containing virus (105 CCID50) (lines 342), and found virus log reduction when treating with for S7, S8, and S9, respectively - 5.75 log, 5.5 log, and 5.5 log for HSV1, and 5.25 log , 5.0 and 5.0 and KHV respectively. Explain what were the titres of virus controls after incubation.

Response 10: Thank you for the note. It should have been 106 CCID50, not 105 CCID50. The correction in the text has been made. The virus titer of the controls is 106.0 CCID50/ml.

Comments 11: Table 5. Pretreatment of MDBK cells with LAB-derived fragments or postmetabolites before viral infection. The title? Did not you pretreat CCB for KHV? What were the titres of virus controls ?

Response 11: Thank you for the remark. Of course, the CCB cells were also pretreated - the results are presented in the table, we inadvertently made an omission to indicate it in the title of the table. We have made the correction. The titer of the virus control is 103 CCID50/ml. Thank you for the question about the titer of the virus control. So we noticed that we have two wrong values ​​in the table, which instead of 3.25 are 3.0. The correction is made in the text. Because of this error, we have reviewed all the results from all the tables for similar errors and have not found others.

Comments 12: Materials and Methods (4.1. Lactobacilli and post-metabolites tested:). Explain how each strain was pre-cultured twice. How many mL of MRS broth was used, what was the temperature for the cultivation, and what was the bacteria dose for the first and the second pre-culturing?

Response 12: Long-term studies of lactic acid bacteria in our laboratory have shown that after storage in a medium with glycerol at minus 20°C, lactobacilli need at least two recultivations to recover their metabolic activity. In this regard, we apply a developed protocol with a two-fold reseeding in liquid MRS broth (10 ml) and a high initial inoculum- 5% v/v, from the previous culture when obtaining postbiotics. The details were added in the section Matheriale & methods. All strains were cultured at 37oC in Incubator Nuve EN 400.

Comments 13: Was the dose the same for every strain? Was the number of bacteria determined after the first and the second preculturing?

Response 13: We tested 4 strains belonging to the species Lactiplantibacillus plantarum and two other Ligilactobacillus salivarius and Lactiplantibacillus paraplantarum. There is no necessity to determine number of bacterial cells at start point of experiments. All stocks made are with high cell density >10 9CFU/mL. Therefore, the second culture, which is the producer is only standardized, (for each strains) according to MakFarland standards. Moreover, we respect the same protocol in preparation starting with 5% initial inoculum from frozen cultures, inoculated to sterile MRS broth. It was optimized to obtain cultures with uniform cell density (reported by McFarland standards). Classical quantitative microbiological analysis (by CFU/mL) is not applicable to the experiment, thus performed for the laboratory production of postmetabolites. To realize production under uniform conditions, all producer strains were standardized to McFarland standards, as was presented in our manuscript.

Comments 14: Lines 262-263 and 28-270 Compare quotes “In order to simulate a multibacterial probiotic formula 4 Lactobacillus strains were combined and mixed CFS were assessed.“ and “Fermented over night milk with a mix of four strains were centrifuged at 1500 rcf 5 min, (Hermle- Germany centrifuge) and the whey fraction (Mix LS-WF) was collected for the assay.” At first, I understood that Lactobacillus strains were not mixed and were not grown together,- and that only CFS of separate strains were mixed, and later it became unclear if milk with a mix of four strains was fermented or not.

Response 14: Thank you for your question. The section was edited carefully:

In order to simulate a multi-bacterial probiotic formula 4 Lactobacillus strains were combined. With this aim initially was obtained four exponential Lactobacillus cultures, with high cell density (4 MacFarland). They are mixed 1:1:1:1 v/v to obtain the inoculum for mixed fermentation. Prepared mixed bacterial culture was inoculated as 5% v/v/ (1.25% v/v from each strain) in 50 ml MRS broth (HiMedia). Obtained CFS after 48 h cultivation at 37oC were assessed.

Harvested cells from the second exponential Lactobacillus cultures, washed with sterile PBS (pH 6.0), were used as inoculum (10% v/v) in sterile reconstituted (10% w/v) skimmed milk (Humana, Germany). Thus, a mixed starter with equal presentation (1:1:1:1 v/v) of each of selected 4 lactobacilli) was prepared and used to inoculate (10% v/v) in sterile milk. Fermented over- night milk at 37oC, with a mixed four Lactobacillus strains, were centrifuged at 1500 rcf 5 min, (Hermle- Germany centrifuge) and the (WF) whey fraction (Mix LS-WF) was collected for the assay. 

Comments 15: 4.4. Cytotoxicity Assay. Line 302 check the incubation time 96 or 120 hours?

Response 15: The two types of cells were incubated for different periods of time, depending on the time interval at which the antiviral experiment was determined (to exclude potential cytotoxicity of the product and to report only its antiviral activity). We have made a correction to the text clarifying which cell line was incubated for which period of time.

Comments 16: Table Lactiplantibacillus but not Lactiplantibaillus

Response 16: The correction has been made.

Comments 17: Choose postmetabolites or post-metabolites

Response 17: Thank you very much. The form postmetabolites was preferable and were edited overall in the text.

Comments 18: 4.5. Determination of Infectious Viral Titers. Lines 320-321 Quote“Viral titers are expressed as lg IU (infectious units) cell culture infectious dose of 50% (CCID50) / 0.1 ml.” 0.1 mL or 1 ml?

Response 18: Viral titers are expressed as lg IU (infectious units) cell culture infectious dose of 50% (CCID50) / 1 ml.

Comments 19: 4.6. Antiviral Activity Assay. Change the title of the section since 4.7, 4.8, and 4.8 mean testing of antiviral activity.

Response 19: The section title was changed. 

4.7. Virucidal Assay

Comments 20: Quote “Samples containing virus (105 CCID50) and tested postmetabolite at its maximum tolerated concentration (MTC) were combined in a 1:1 ratio”. What was the volume? Explain more in detail. What did the volume of the sample contained 105 CCID50?

Response 20: The sample to be prepared is 1 ml in volume. 500 µL of the virus suspension and 500 µL of 2MTC of the respective postmetabolite. The sample is incubated for the indicated time intervals, after which descending dilutions are made from each sample and the decrease in the viral titer is determined.

Comments 21: 70% ethanol control is not described.

Response 21: The use of 70% ethanol as a control in the experiment has been added to the text.

Comments 22: 4.9. Pre-Treatment of Healthy Cells. What about pretreatment of CCB for KHV testing?

Response 22: Pretreatment of CCB cells was performed as for MDBK cells. Information is supplemented in the text.

Reviewer 2 Report

Comments and Suggestions for Authors

Lactobacilli-Derived Postmetabolites As Broad-Spectrum Inhibitors Of Herpes Viruses In Vitro

The authors have identified several lactic acid bacteria metabolites that can reduce HSV-1 and KHV virus replication in MDBK cells and CCB cells in terms of post and pre treatment conditions.

Major comments

1. The authors should examine the composition of each metabolite, as it represents a key finding of this study.

2. For Tables 1 and 2, the authors should include cytotoxicity and inhibitory curves (graphs) for all tested samples across both cell types. Additionally, clarify which cell line was used for the IC50 assay.

3.Table 1-lines 96-97 – CC50 is the testing compound at which reduces the cell viability by 50%. In that case lines 96-97 is misleading in which ACV is less toxic than tested samples. Please revise the sentence.

4. It is expected that except S1 and S2, other metabolites reduce the virus attachment since those metabolites deployed virucidal activity. In this term, why the authors have conducted attachment inhibition assay for S4-S9 metabolites. And in the post treatment assay what are the conclusions for tested metabolites on how they inhibit virus replication, whether metabolites’ virucidal activity or attachment inhibition.

5. The authors have identified that pre-treatment with metabolites inhibits virus replication. They must demonstrate the mechanism by which these metabolites block virus replication. This can include determining how the metabolite treatments activate the innate immune system.

Minor comments

1. To enhance understanding and readability, please provide plotted graphs for the data presented in Tables 3, 4, and 5.

2. Its not clear how the authors prepared S1-S9 metabolites in the materials and methods section. Please revise the relevant section for better understanding of sample preparation specially lines 267 to 270.

Author Response

Major comments

Comments 1: The authors should examine the composition of each metabolite, as it represents a key finding of this study.

Response 1: Thank you very much for such important idea. We just starting with detailed metabolomics analyses, by gas chromatography and 49 polar and non-polar compounds have been identify as percent from the sample. Thus, a quantitative estimation of each of them is not possible, due to the strain-specificity and dynamic of production in a specific food matrix for each Lactobacillus strains. A future work will be presented and detailed results with more information in relation with genetic base of mode of action.

Comments 2: For Tables 1 and 2, the authors should include cytotoxicity and inhibitory curves (graphs) for all tested samples across both cell types. Additionally, clarify which cell line was used for the IC50 assay.

Response 2: For Tables 1 and 2, the inhibitory curves for all tested samples are included in the manuscript. The IC50 was determined for each virus strain on the cell line on which the respective virus strain was cultured and on which subsequent studies were performed. This is indicated in Materials and Methods in section 4.3. For HSV-1, this is the MDBK cell line, and for KHV, CCV cells.

Comments 3: Table 1-lines 96-97 – CC50 is the testing compound at which reduces the cell viability by 50%. In that case lines 96-97 is misleading in which ACV is less toxic than tested samples. Please revise the sentence.

Response 3: The sentence has been revised.

Comments 4: It is expected that except S1 and S2, other metabolites reduce the virus attachment since those metabolites deployed virucidal activity. In this term, why the authors have conducted attachment inhibition assay for S4-S9 metabolites. And in the post treatment assay what are the conclusions for tested metabolites on how they inhibit virus replication, whether metabolites’ virucidal activity or attachment inhibition.

Response 4: The effect on extracellular virions and the effect on their adsorption stage to susceptible cells are two different stages of the existence and reproduction of the virus. Sometimes there are substances with virucidal activity that do not necessarily have an effect on the stage of viral adsorption and vice versa. Therefore, we have examined all nine post-metabolites for any possibility of antiviral activity. We do not exclude the possibility that the effect on the stage of viral adsorption is largely due to virucidal action. However, our results show a significantly stronger effect in blocking the adsorption stage compared to the virucidal activity. This indicates that in addition to the interaction between the components of the postmetabolites and the viral particles, there is most likely also an interaction between the components of the studied samples and the cellular receptors responsible for the attachment and penetration of the virus. If some of the samples were not examined for their influence on the viral adsorption stage, we would not have established their remarkable inhibitory effect, which was reported by Δlg ≥ 5. As the experiment was carried out with the maximum non-toxic concentration of the products. This means that the corresponding influence on the cells does not affect their viability.

This is commented on in the discussion of the manuscript.

Comments 5: The authors have identified that pre-treatment with metabolites inhibits virus replication. They must demonstrate the mechanism by which these metabolites block virus replication. This can include determining how the metabolite treatments activate the innate immune system.

Response 5: Pre-treatment of cells with post-metabolites most likely reduces the amount of attached and penetrated viral particles. This method of treatment does not affect the stage of viral replication in the cell, because when the viral replicative cycle begins, the studied products have already been removed from the cells. Most likely, the effect is due to interaction with cellular receptors responsible for the attachment and penetration of the virus into the cell. The same effect that we assume causes the strong effect reported when influencing the stage of viral adsorption. It is possible that in this way the body's protective barriers for insensitivity to the given viral infections can be activated. However, we cannot demonstrate this study at the moment. For its implementation, it is more appropriate to conduct in vivo experiments (which are foreseen in our future plans), but at the moment we do not have the necessary permits to conduct them.

Minor comments

Comments 1: To enhance understanding and readability, please provide plotted graphs for the data presented in Tables 3, 4, and 5.

Response 1: Thank you for the suggestion. The manuscript includes plotted graphs of the data presented in Tables 3, 4, and 5.

Comments 2: Its not clear how the authors prepared S1-S9 metabolites in the materials and methods section. Please revise the relevant section for better understanding of sample preparation specially lines 267 to 270.

Response 2: Thank you very much, The necessary information was added as follow:

For experimental purposes, each strain was pre-cultured twice in MRS broth (HiMedia, Bombay, India) in order to obtain high number of living cells synchronized, as growth phase and metabolic activity. The BLIS production is quorum sensing process. Thus, after storage at (-20oC) in medium supplemented with glycerol (20% v/v) two passage in fresh MRS broth were needed. A 5% inoculum (v/v) from a Lactobacillus’ stock was used in the first culture for each of selected Lactobacilli. They were incubated at 37oC and obtained late exponential phase cultures, were estandardized by MacFarland 4.0 standards (~1.2 x109 CFU/mL) were used as 10% (v/v) inoculum for second cultures. The cell-free supernatants (CFS) were collected by centrifugation (3000 rcf, 5 min, Hermle- Germany centrifuge) and were used for in vitro test.

Round 2

Reviewer 1 Report

Comments and Suggestions for Authors

The article needs several corrections

 Read https://ictv.global/faq/names and https://ictv.global/faq/names and follow the rules when you need to italicize and when you do not

Lines 46-47. Use Italic and write Alloherpesviridae, Orthoherpesviridae and Malacoherpesviridae. Finish the sentence with point (.).

Lines 46, 50 Orthoherpesviridae but not Herpesviridae - read https://ictv.global/taxonomy

Lines 63, 70, 111, 226, 227, 231, 242, 276, and 419 - the word “Lactobacilli” does not have to be italicized.

Lines 108, 133, and 276  - you had chosen to write postmetabolites but still use post-metabolites

Lines 289, 302 „Hermle- Germany”? Correct

Lines 289 Can you confirm that you used the broth named „De Man Rogosa”? Is it the correct name of the broth? You have to write the producer of the broth.

You have to write °C – take a look at lines 285 and 287.

Author Response

We thank the esteemed reviewer for his recommendations and remarks, which led to an improvement in the quality of our manuscript.

Comments 1: Read https://ictv.global/faq/names and https://ictv.global/faq/names and follow the rules when you need to italicize and when you do not

Response 1: Thank you for the recommendation.

Comments 2: Lines 46-47. Use Italic and write Alloherpesviridae, Orthoherpesviridae and Malacoherpesviridae. Finish the sentence with point (.).

Response 2: The corrections have been made.

Comments 3: Lines 46, 50 Orthoherpesviridae but not Herpesviridae - read https://ictv.global/taxonomy

Response 3: The correction have been made. 

Comments 4: Lines 63, 70, 111, 226, 227, 231, 242, 276, and 419 - the word “Lactobacilli” does not have to be italicized.

Response 4: The corrections have been made, except for paragraph 276, because there is part of a subsection title that is in italics.

Comments 5: Lines 108, 133, and 276  - you had chosen to write postmetabolites but still use post-metabolites

Response 5: The corrections have been made.

Comments 6: Lines 289, 302 „Hermle- Germany”? Correct

Response 6: Information about the manufacturer has been added to the text.

Comments 7: Lines 289 Can you confirm that you used the broth named „De Man Rogosa”? Is it the correct name of the broth? You have to write the producer of the broth.

Response 7: The full name of the broth has been added and the manufacturer has been listed.

Comments 8: You have to write °C – take a look at lines 285 and 287.

Response 8: The corrections have been made.

Reviewer 2 Report

Comments and Suggestions for Authors

Lactobacilli-Derived Postmetabolites As Broad-Spectrum Inhibitors Of Herpes Viruses In Vitro

Major comments

This reviewer is satisfied, except for the response to comment 3. The authors have not addressed comment 3 raised by this reviewer.

Author Response

Comments 1: This reviewer is satisfied, except for the response to comment 3. The authors have not addressed comment 3 raised by this reviewer.

Response 1: We thank the esteemed reviewer for the recommendations and comments made so far. We are convinced that they helped to improve the quality of our manuscript.

The first time you made comment 3 to us, we made a correction to the text. For some reason, it was not saved, along with other corrections that we had made to the text. Fortunately, we now have the opportunity to make the corrections again. Thank you again for the repeated comment.